# Effect of Limb-Specific Resistance Training on Central and Peripheral Artery Stiffness in Young Adults: A Pilot Study

**Minyoung Kim [1], Ruda Lee [1], Nyeonju Kang [2,3] and Moon-Hyon Hwang [3,4,*]**

[1] Department of Human Movement Science, Graduate School, Incheon National University, Incheon 22012, Korea; sks911222@naver.com (M.K.); winner72@inu.ac.kr (R.L.)
[2] Division of Sport Science, Incheon National University, Incheon 22012, Korea; nyunju@inu.ac.kr
[3] Sport Science Institute, Incheon National University, Incheon 22012, Korea
[4] Division of Health & Kinesiology, Incheon National University, Incheon 22012, Korea
[*] Correspondence: mhwang@inu.ac.kr; Tel.: +82-32-835-8698

**Abstract:** This study aimed to investigate the effect of limb-specific resistance training on arterial stiffness in young adults. Twenty-four participants were randomly assigned to three groups: upper-limb resistance training ($n = 8$ (URT)), lower-limb resistance training ($n = 8$ (LRT)), and control group ($n = 8$ (CON)). Both URT and LRT groups performed the limb-specific resistance training at 70–80% of one-repetition maximum twice a week for 8 weeks. The aortic pulse wave velocity and augmentation index (AIx) were measured by the SphygmoCor XCEL to assess central artery stiffness. Peripheral artery stiffness was evaluated by brachial to radial artery pulse wave velocity (ArmPWV) and femoral to posterior tibial artery pulse wave velocity (LegPWV) using Doppler flowmeters. URT significantly reduced AIx ($4.7 \pm 3.0$ vs. $0.3 \pm 2.9\%$, pre vs. post, $P = 0.01$), and ArmPWV presented a tendency to decrease following URT ($10.4 \pm 0.3$ vs. $8.6 \pm 0.8$ m/s, pre vs. post, $P = 0.06$). LRT showed no negative influence on central and peripheral artery stiffness. Changes in serum triglyceride and leg lean body mass after resistance training were significantly associated with changes in AIx and LegPWV, respectively. URT is beneficial in decreasing central artery wave reflection and may help to improve local peripheral artery stiffness even in healthy young adults.

**Keywords:** resistance training; arterial stiffness; pulse wave velocity; augmentation index

## 1. Introduction

Cardiovascular disease (CVD) is the leading cause of death, and arterial stiffness is an early marker of CVD risk [1]. Large elastic arteries buffer augmented pulsatile energy and blood pressure when the heart pumps blood into the systemic vascular network [2]. Thus, an increase in large elastic artery stiffness augments central blood pressure and left ventricular load, which in turn reduces coronary artery perfusion and may increase acute cardiac event risk [3,4]. Even in young adults, a decrease in arterial distensibility, a measure of arterial stiffness, increases the number of cardiovascular risk factors [5].

Both aerobic and resistance exercises are recommended to prevent chronic diseases and to promote overall health in various populations [6–12]. Regular aerobic exercise is a well-known intervention to reduce blood pressure and arterial stiffness [13–15]. Resistance exercise is a typical physical activity and is commonly prescribed to enhance musculoskeletal and cardiovascular function [16,17]. Compared to its superior effect on musculoskeletal function and metabolic efficiency, the effect of resistance training on cardiovascular function, particularly arterial stiffness, is still controversial. Miyachi et al. reported that long-term resistance training decreased arterial stiffness in young and middle-aged men [18], but another previous finding demonstrated that resistance training impaired arterial stiffness, including carotid artery compliance [19]. Regarding training intensity, low-intensity resistance exercise decreased arterial stiffness, whereas moderate- to high-intensity resistance exercise augmented arterial stiffness in healthy young adults [20–24].

Furthermore, previous studies have hardly investigated thoroughly the benefit of limb-specific resistance exercise on both central and peripheral artery stiffness with validated vascular size-specific (central vs. peripheral) research methods. Thus, the purpose of this study was to examine the effect of limb-specific resistance training on both central and peripheral artery stiffness in healthy young adults.

## 2. Materials and Methods

### 2.1. Participants

Twenty-five young adults volunteered to participate in this study. They had not been doing any regular resistance training (that is, any type of resistance exercise for more than 2 days a week) for at least 6 months. Participants were excluded from this study if they had a smoking history in the last 5 years; any overt clinical diseases, including CVD, diabetes, and metabolic syndrome; or musculoskeletal problems that limit resistance exercises. A total of 24 participants (9 men and 15 women; 18–25 years old) finished this study, except for 1 participant who sustained a muscular injury unrelated to this study. Each participant voluntarily signed an informed consent after a thorough explanation of the nature, purposes, and risks of the study. The Institutional Review Board of Incheon National University approved this study. The study was conducted in accordance with the ethical standards of the Declaration of Helsinki.

### 2.2. Study Design

The study participants were randomly assigned to three groups: nonexercise control ((CON) $n = 8$), upper-limb resistance training ((URT) $n = 8$), and lower-limb resistance training ((LRT) $n = 8$). The participants in the CON group were asked to maintain their normal lifestyle for 8 weeks. The young adults in the URT and LRT groups were required to maintain their normal lifestyle and to complete the scheduled resistance exercise sessions. At both pre- and post-training, physiological parameters were obtained in a supine position in a temperature-controlled and semidarkened room and taken at the same time in the morning after at least 10 h of an overnight fast. To minimize the acute effect of resistance exercise, post-training measures were performed 20 to 24 h after finishing the last exercise session in the training groups. All physiological measures on female participants were performed in the early follicular phase to exclude the confounding effects of sex hormones on vascular function. All research procedures and supervised resistance exercise sessions were implemented in the Exercise & Cardiovascular Physiology Laboratory at Incheon National University.

### 2.3. Height, Weight, and Body Composition

The participants' height was measured in mm using a stadiometer. Body weight, fat mass, percent body fat, body mass index, and lean body mass (LBM) were evaluated by a segmental bioelectrical impedance analysis device (Inbody 720, Biospace, Seoul, Korea).

### 2.4. Central Artery Stiffness

Augmentation index (AIx) and aortic pulse wave velocity (AorPWV), which are validated, noninvasive measures of arterial stiffness, were evaluated by the SphygmoCor XCEL system (AtCor Medical, Sydney, Australia) [25–28]. The participants rested in a supine position for 15 min prior to the measurements following at least 10 h of overnight fast. To measure AIx, brachial artery pressure waveforms were obtained by a blood pressure cuff placed on the right upper arm. The brachial waveforms were then automatically transformed into central artery waveforms by the mathematical transfer function embedded in the system. The system estimated AIx as the ratio of amplified systolic blood pressure to pulse pressure in the central artery. AorPWV, which is a reliable, noninvasive measure of arterial stiffness in humans, was assessed by simultaneously measuring the carotid pulse waveforms by applanation tonometry and the femoral pulse waveforms by a blood pressure cuff placed on the right upper leg proximal to the inguinal ligament [28,29]. In

the system, AorPWV was automatically calculated as the ratio of the distance between the two arterial measuring sites and the time of pulse waves moving between these two sites. The average of the three high-quality measures was used for AIx and AorPWV. The distance between the carotid and femoral artery measuring site was defined as previously mentioned [28]. In our laboratory, the day-to-day intertest coefficients of variation for AorPWV and AIx measurements were 4.4% and 11.2%, respectively.

### 2.5. Peripheral Artery Stiffness

A transcutaneous Doppler flowmeter (model 810-A, Parks Medical Electronics, Inc., Aloha, OR, USA) was employed to measure arm pulse wave velocity (ArmPWV) and leg pulse wave velocity (LegPWV). Pressure waves measured at the peripheral artery sites were digitized with a signal processing data acquisition system (model PL2604, AD Instruments Inc., Colorado Springs, CO, USA). Through an offline analysis, PWV was calculated as the distance in meters divided by the pulse transit time in seconds between the two recording sites. ArmPWV and LegPWV were determined by taking the average of the PWVs obtained from 10 paired pressure waveforms in the brachial and radial arteries and in the femoral and posterior tibial arteries, respectively. Due to the technical difficulty in measuring the high-quality arterial pressure waveforms at the two recording sites at the same time, electrocardiogram R-waves simultaneously measured with the pressure waveforms were used as a reference mark when calculating the pulse transit time between the two recording sites.

### 2.6. Blood Chemistry

To analyze the effects of resistance training on traditional CVD risk, serum triglyceride was analyzed by the enzymatic colorimetric assay using a triglycerides assay kit (TRIGL, Roche, Germany) and the Cobas 8000 analyzer (c702, Roche, Germany) pre- and post-training. The blood concentrations of epinephrine and norepinephrine were assessed by high-performance liquid chromatography (Acclaim, Bio-Rad, Hercules, CA, USA) using a plasma catecholamines assay kit (Plasma Catecholamines by HPLC, Bio-Rad, Feldkirchen, Germany) to investigate changes in the autonomic nervous system function and related hormones after the scheduled resistance exercise sessions. All of the blood chemistries were performed within a clinical laboratory. To avoid the influence of invasive blood from drawing on other physiological measures, the blood collection was performed following other physiological measures pre- and post-training.

### 2.7. Maximal Strength

The maximal strength of one-repetition maximum (1 RM) for chest and shoulder press, seated row, and barbell curl exercise was measured during URT, and 1 RM for leg extension, leg press, lying leg curl, and hip extension exercise was evaluated for LRT. In this study, 1 RM for each exercise was determined by an indirect estimation equation after pre- and post-training measures to maximize safety as previously described [30]. To summarize, the participants performed warm-up and stretching exercises for 5 min prior to the test. The warm-up exercise was composed of resistance exercises with a load of about 10 RM. In the test, the weight load was progressively increased to 2–5 RM load with a 3-min rest after beginning the test with 8–10 RM load. The indirect estimation equation for 1 RM was presented as W0 + W1. In this equation, W0 indicates the weight load considered to be slightly heavy by the participants between 2 and 8 RM load, and W1 means the following calculation: W0 $\times$ 0.025 $\times$ the number of repetitions.

### 2.8. Resistance Training Program

URT and LRT groups performed the scheduled resistance exercise sessions 2 days a week (either Monday and Wednesday or Tuesday and Thursday) for 8 weeks using an air resistance weight machine system (HUR Oy, Kokkola, Finland). The resistance training program was established to perform 4 exercises per session, 5 sets per exercise, and

10 repetitions per set with 2 min of rest between sets and exercises. The resistance training program for URT consisted of chest and shoulder press, seated row, and barbell curl exercises. Leg extension, lying curl, leg press, and hip extension exercises were included in the LRT program. Exercise intensity was set between 70 and 80% of 1 RM in each resistance exercise; the intensity between the 1st and 4th week was established at 70% of 1 RM and was adjusted to 80% of 1 RM thereafter. All of the exercise sessions were supervised by an exercise physiologist who provided verbal motivation and feedback.

### 2.9. Statistical Analyses

Statistical analyses were performed using an SPSS Statistics program (version 24, IBM SPSS Inc., Armonk, NY, USA). A one-way analysis of variance was used to examine whether pre-intervention group differences existed or not. In each group, the effect of intervention on the main dependent variables was evaluated by a paired t-test. Relationships between changes in arterial stiffness and other physiological variables were assessed using Pearson's correlation coefficient because all data included in this correlation analysis were normally distributed. Statistical significance was set at $P < 0.05$.

### 3. Results

The participants' characteristics are presented in Table 1.

**Table 1.** Participant characteristics pre- and post-intervention.

| Variables | CON (*n* = 8) | | URT (*n* = 8) | | LRT (*n* = 8) | |
|---|---|---|---|---|---|---|
| | Pre | Post | Pre | Post | Pre | Post |
| Age, years | 21 ± 1 | - | 20 ± 1 | - | 20 ± 1 | - |
| Height, cm | 164.4 ± 1.2 | - | 172.8 ± 4.7 | - | 170.2 ± 2.0 | - |
| Weight, kg | 60.0 ± 3.3 | 59.0 ± 3.5 | 65.0 ± 4.9 | 67.6 ± 5.7 | 65.1 ± 3.6 | 63.4 ± 4.0 |
| BMI, kg/m² | 21.8 ± 1.1 | 21.8 ± 1.2 | 21.5 ± 0.6 | 21.9 ± 0.7 | 22.4 ± 0.9 | 22.7 ± 1.0 |
| Body fat, % | 23.1 ± 2.5 | 23.9 ± 2.3 | 18.5 ± 2.4 | 17.4 ± 2.5 | 24.3 ± 2.2 | 25.8 ± 2.4 |
| Muscle mass, kg | 25.0 ± 1.3 | 24.8 ± 1.2 | 30.2 ± 3.4 | 31.0 ± 3.4 * | 27.6 ± 2.2 | 28.1 ± 2.1 |
| Trunk LBM, kg | 20.0 ± 1.0 | 19.7 ± 0.9 | 22.4 ± 2.0 | 23.2 ± 2.2 * | 20.9 ± 1.4 | 21.3 ± 1.4 * |
| Arm LBM, kg | 2.3 ± 0.2 | 2.2 ± 0.2 | 2.6 ± 0.3 | 2.8 ± 0.4 * | 2.4 ± 0.2 | 2.4 ± 0.2 * |
| Leg LBM, kg | 7.0 ± 0.3 | 7.1 ± 0.3 | 8.7 ± 1.1 | 8.8 ± 1.0 | 8.0 ± 0.6 | 8.1 ± 0.6 |
| rHR, beat/min | 59 ± 3 | 58 ± 2 | 58 ± 3 | 52 ± 2 * | 57 ± 3 | 53 ± 3 |
| SBP, mmHg | 110 ± 2 | 111 ± 4 | 115 ± 4 | 114 ± 4 | 113 ± 3 | 109 ± 4 |
| DBP, mmHg | 64 ± 2 | 64 ± 2 | 64 ± 2 | 61 ± 3 | 63 ± 1 | 63 ± 3 |
| Triglyc, mg/dL | 64 ± 6 | 74 ± 13 | 66 ± 8 | 67 ± 7 | 71 ± 11 | 72 ± 12 |
| Epi, pg/mL | 43 ± 2 | 35 ± 3 | 41 ± 2 | 32 ± 3 | 52 ± 7 | 34 ± 3 |
| Norepi, pg/mL | 325 ± 33 | 128 ± 14 * | 413 ± 36 | 140 ± 15 * | 372 ± 25 | 118 ± 19 * |

Data are mean ± SE. Abbreviations: CON, nontraining control group; URT, upper-limb resistance training group; LRT, lower-limb resistance training group; BMI, body mass index; LBM, lean body mass; rHR, resting heart rate; SBP, systolic blood pressure; DBP, diastolic blood pressure; Triglyc, triglycerides; Epi, epinephrine; Norepi, norepinephrine. There was no significant group difference at pre-intervention. * $P \leq 0.05$ vs. pre-intervention.

No significant difference was noted in age, weight, heart rate and blood pressure, and blood lipid and catecholamine levels among the CON, URT, and LRT groups prior to the intervention ($P \geq 0.16$; Table 1). No significant difference was observed in body composition parameters, including weight, body mass index, percent body fat, muscle mass, and segmental (trunk, arm, and leg) LBM, among the three groups at baseline ($P \geq 0.21$; Table 1). As expected, the abovementioned body composition factors did not change in the CON group after maintaining a nonexercise lifestyle during the intervention. The URT group had a significant improvement in muscle mass and arm LBM ($P \leq 0.01$); however, the LRT group showed no improvement in muscle mass and leg LBM after the established intervention ($P \geq 0.12$; Table 1). An increased trunk LBM was observed in the URT and LRT groups following the training ($P \leq 0.02$; Table 1). As a result of resistance training, the maximal strength of every exercise in both URT and LRT groups significantly improved between 14.0 and 24.3% ($P \leq 0.006$; Table 2).

**Table 2.** Changes in maximal strength (1 RM) after resistance training.

| Group | Exercise | Pre-Intervention | Post-Intervention | *t*-Test *P* Value | Δ1 RM (%) |
|---|---|---|---|---|---|
| URT | Chest press, kg | 66.8 ± 3.0 | 81.7 ± 2.9 | 0.0001 | 22.3 |
| | Shoulder press, kg | 58.5 ± 8.4 | 71.2 ± 11.1 | 0.006 | 21.7 |
| | Seated row, kg | 20.2 ± 1.2 | 23.3 ± 0.5 | 0.004 | 15.3 |
| | Barbell curl, kg | 24.3 ± 2.8 | 30.2 ± 2.7 | 0.0001 | 24.3 |
| LRT | Leg press, kg | 182.7 ± 14.0 | 208.3 ± 14.8 | 0.001 | 14.0 |
| | Leg extension, kg | 78.4 ± 5.6 | 93.8 ± 4.3 | 0.0001 | 19.6 |
| | Lying leg curl, kg | 15.4 ± 1.7 | 18.8 ± 1.7 | 0.0001 | 22.1 |
| | Hip extension, kg | 13.9 ± 1.7 | 16.4 ± 1.7 | 0.0001 | 18.0 |

Data are mean ± SE. Abbreviations: RM, repetition maximum; URT, upper-limb resistance training group; LRT, lower-limb resistance training group.

In response to the 8-week intervention, AIx was significantly reduced in the URT group ($P$ = 0.01; Figure 1A), but no change was noted in the LRT and CON groups. URT and LRT did not lead to a significant change in AorPWV compared to the baseline value (Figure 1B). ArmPWV showed a tendency to decrease following URT, but this tendency did not reach statistical significance ($P$ = 0.06; Figure 1C). No change was observed in the LegPWV following LRT (Figure 1D).

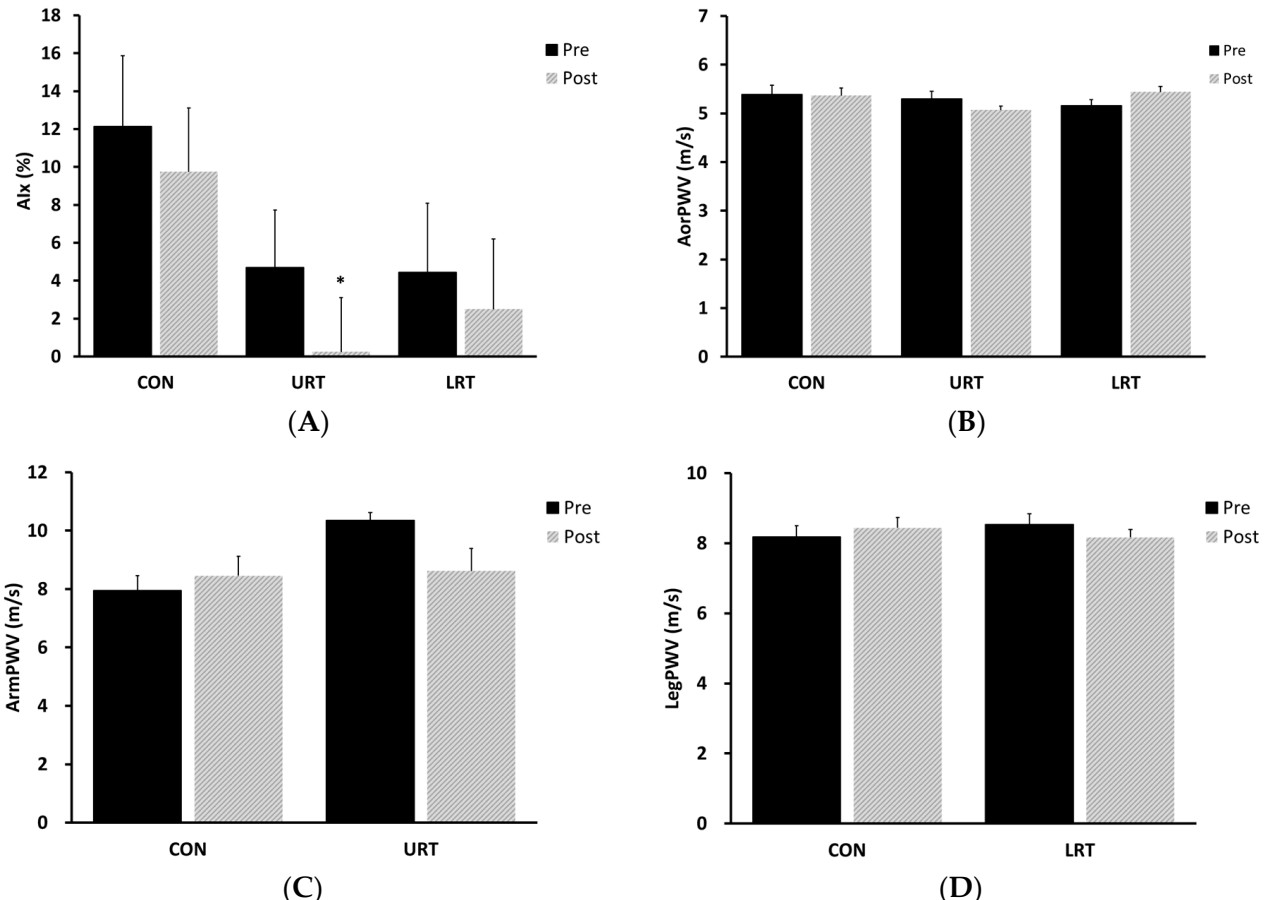

**Figure 1.** Change in central (AIx and AorPWV, panels (**A**) and (**B**), respectively) and peripheral (ArmPWV and LegPWV, panels (**C**) and (**D**), respectively) artery stiffness in response to the intervention. AIx, augmentation index; AorPWV, aortic (carotid–femoral artery) pulse wave velocity; ArmPWV, arm (brachial–radial artery) pulse wave velocity, LegPWV, leg (femoral–posterior tibial artery) pulse wave velocity; CON, nontraining control group; URT, upper-limb resistance training group; LRT, lower-limb resistance training group.

Plasma epinephrine concentration tended to decrease in the three groups but showed no statistical significance ($P \geq 0.07$; Table 1). The norepinephrine level was significantly reduced in all groups ($P \leq 0.001$; Table 1). In the resistance training groups, changes in AIx and LegPWV were significantly associated with changes in serum triglyceride levels and leg LBM, respectively (r = 0.57 and −0.61, $P$ = 0.02 and 0.01; Figure 2). The changes in AorPWV and trunk LBM in response to resistance training seemed related to each other but did not reach statistical significance (r = 0.41, $P$ = 0.06).

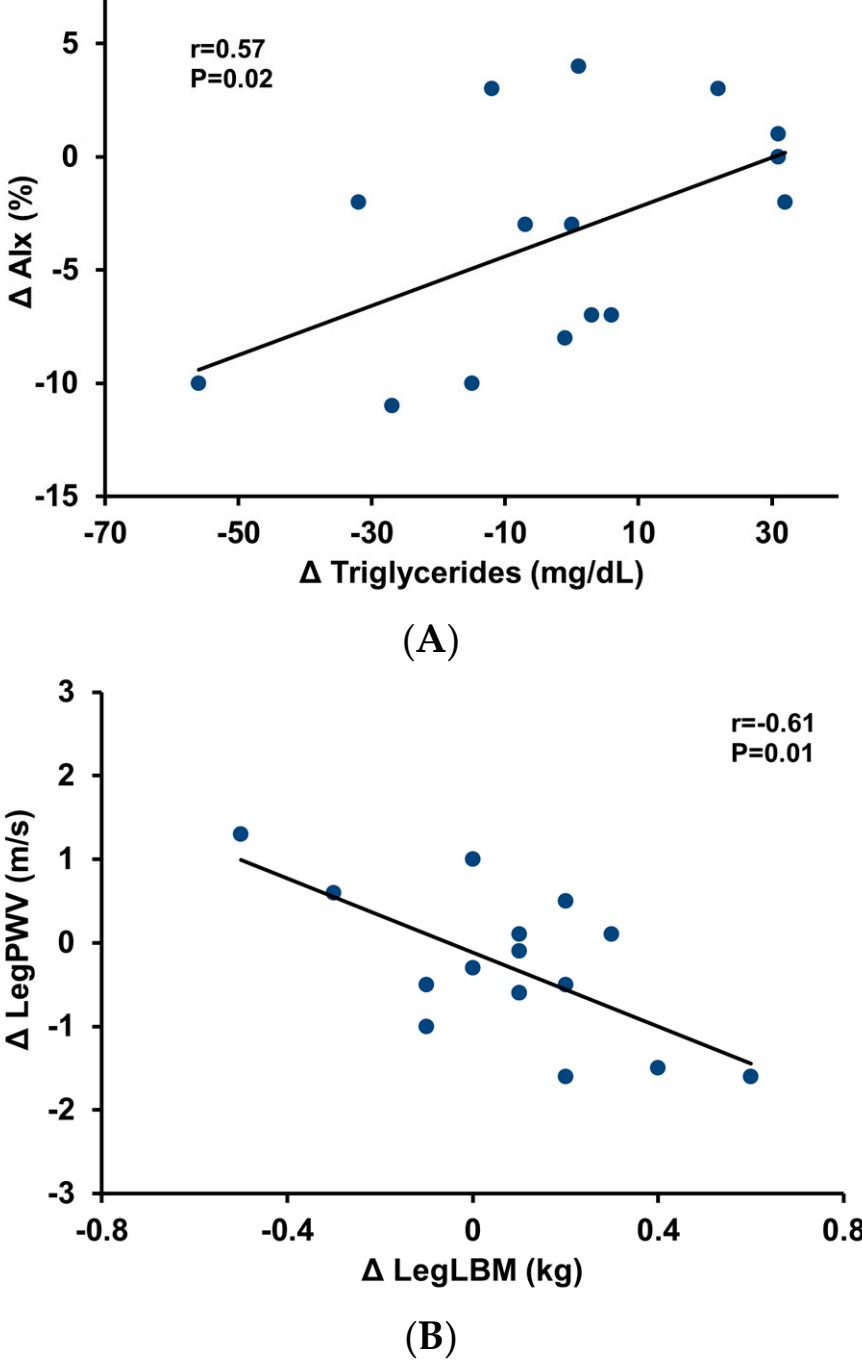

**Figure 2.** Relationship between the change in triglycerides and the change in augmentation index in response to resistance training (panel (**A**)). Relationship between the change in leg lean body mass and the change in leg pulse wave velocity in response to resistance training (panel (**B**)). AIx, augmentation index; PWV, pulse wave velocity; LBM, lean body mass.

## 4. Discussion

This study was performed to investigate the effect of limb-specific resistance training on both central and peripheral artery stiffness in young adults. It is, to our knowledge, the first to assess the effect of limb-specific long-term resistance training on both central and peripheral artery stiffness in young adults without any clinical disease.

Resistance exercise is one type of exercise recommended for improving muscular strength and cardiovascular function and for preventing musculoskeletal diseases, such as osteopenia and osteoporosis [31]. Although resistance exercise is recommended to enhance cardiovascular health, the effect of resistance exercise on arterial stiffness is still controversial. In this study, 8 weeks of URT reduced AIx, but not AorPWV, in young adults. For the same duration, LRT did not change the central artery stiffness measures. It has been documented that change in the resting heart rate after exercise training influences AIx, which may result from the adaptation of the autonomic nervous system to exercise training [32,33]. In this study, the decrease in the resting heart rate and the systemic norepinephrine concentration (Table 1) following URT may in part explain the reduction in AIx. Furthermore, previous studies imply that an increase in muscular strength after exercise training is associated with a decrease in central artery stiffness. Fahs et al. reported that adults who have higher muscular strength present lower central artery stiffness; particularly, those who have higher upper-limb muscular strength tend to have lower central artery stiffness [34]. In response to 8 weeks of URT, 1 RM of upper-limb resistance exercises was improved with a simultaneous increase in arm LBM in this study. The increase in muscular strength accompanying muscle hypertrophy may in part account for AIx reduction after URT. Change in AIx after the resistance exercise was associated with change in serum triglyceride levels. Blood triglyceride concentration measured after overnight fasting is used as a traditional marker of CVD risk. Considering both that endothelial dysfunction is a strong early marker to predict future CVD risk and that vascular endothelial dysfunction is closely related to serum triglyceride concentration, a decrease in serum triglyceride concentration after resistance training may contribute to a decrease in central artery stiffness via enhanced vascular endothelial function [35].

ArmPWV, a measure of peripheral artery stiffness in this study, showed a tendency to decrease in response to URT, although this tendency was not statistically significant. This result is different from the results of previous studies. Okamoto et al. reported that 8 weeks of single arm-curl resistance exercise increased baPWV, which is another measure of peripheral artery stiffness reflecting mainly lower-limb artery stiffness, in young women [36]. Similarly, 10 weeks of URT increased baPWV in young adults [37]. However, LRT did not influence baPWV in young adults and did not show any negative effects on AorPWV, the gold standard measure of central artery stiffness, in older adults [37,38]. The acute effects of one bout of resistance exercise on arterial stiffness can be different based on the exercising limbs. Li et al. reported that one bout of upper-limb resistance exercise increased baPWV, but lower-limb resistance exercise did not increase central artery stiffness in young men [39]. Similarly, in young adults, one bout of lower-limb resistance exercise did not show any negative effect on central artery stiffness [40]. In previous findings, the effects of resistance exercise on arterial stiffness were different according to the participant's age, gender, exercising limbs, training duration, and vascular beds measured for arterial stiffness assessment. Additionally, it is speculated that any difference in the effect of resistance exercise on peripheral artery stiffness between this study and the previous findings may be due to the difference in the employed methodologies—ArmPWV vs. baPWV.

No improvement was noted in the LegPWV following the 8-week LRT. However, for both resistance training groups (URT and LRT), the change in leg LBM had a significant relationship with the change in LegPWV. Resistance training can reduce peripheral artery stiffness not only by decreasing the trained peripheral artery tone at rest [41,42] but also by local vasodilation induced by increased circulating metabolites generated from resistance training [43]. Thus, this result suggests that improved vascular endothelial function after

limb-specific resistance training via both local and systemic biochemical environment change may lead to enhanced peripheral artery stiffness even in relatively healthier young adults. Unfortunately, the related physiological mechanism could not be demonstrated because vascular endothelial function measures were not performed in this study.

Resting norepinephrine concentration significantly decreased and resting epinephrine concentration showed a tendency to decrease in all three groups following the 8 weeks of intervention. In particular, the reduction of systemic norepinephrine level in CON is likely associated with the impact of seasonal ambient temperature variations on the autonomic nervous system. The plasma catecholamine level in the cold ambient temperature is higher than that in the warm outside temperature, which is likely related to the increased cardiovascular morbidity and mortality in winter [44–49]. In this study, blood collection pre- and post-intervention was performed in March and July in 2017. The average ambient temperatures at Incheon, Korea during March and July in 2017 were 5.8 and 25.8 °C, respectively. It is speculated that the seasonal difference of ambient temperature might cause the significant difference in resting norepinephrine concentration between pre- and post-intervention.

This study has limitations that should be considered when interpreting major findings. The study was conducted with a relatively small number of healthy young adults. Thus, the research results cannot be generalized and applied to other populations with different biological or clinical conditions. In this study, the number of study participants by gender was not able to secure statistical power, making it impossible to further analyze gender differences. This study mainly analyzed functional changes, and there was a limit to the analysis of physiological mechanisms. In the future, large-scale functional and mechanistic research studies in which a sufficient number of men and women of various ages and health conditions participate are needed.

## 5. Conclusions

URT reduces pulse wave reflection, an index of arterial stiffness, and may be beneficial in decreasing local peripheral artery stiffness even in healthy young adults. Decreased serum triglyceride concentration and increased regional muscle mass in response to resistance training may contribute to the reduction of pulse wave reflection and peripheral artery stiffness in young individuals. These findings have clinical implications for the improvement of cardiovascular function and the prevention and management of cardiovascular disease for individuals with limitations in lower-limb movement or activity, such as injured veterans and disabled persons.

**Author Contributions:** Conceptualization, M.K. and M.-H.H.; methodology, M.K., R.L. and M.-H.H.; formal analysis, M.K., N.K. and M.-H.H.; investigation, M.K. and R.L.; resources, M.-H.H.; data curation, N.K. and M.-H.H.; writing—original draft preparation, M.K.; writing—review and editing, R.L., N.K. and M.-H.H.; supervision, M.-H.H.; project administration, M.K. and M.-H.H.; funding acquisition, M.-H.H. All authors have read and agreed to the published version of the manuscript.

**Funding:** This research was funded by the Incheon National University Research Grant in 2016 to Moon-Hyon Hwang.

**Institutional Review Board Statement:** The study was conducted according to the guidelines of the Declaration of Helsinki and approved by the Institutional Review Board of Incheon National University (approval# 7007971-201612-004-01 and the study protocol was approved on 12 December 2016).

**Informed Consent Statement:** Informed consent was obtained from all subjects involved in the study.

**Data Availability Statement:** Not applicable.

**Acknowledgments:** We thank the study participants for their time and effort.

**Conflicts of Interest:** The authors declare no conflict of interest.

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
