# Peer review of "Effect of Limb-Specific Resistance Training on Central and Peripheral Artery Stiffness in Young Adults: A Pilot Study"

_applsci, doi:10.3390/app11062737_

Round 1

Reviewer 1 Report

I am not convinced with some results show of this study, mainly those reported from line 165 to line 186. Furthermore, the number of participants is very low (only 24). The results are not significant and they could only be showed as a pilot experiment. If the paper were accepted, this aspect should show in somewhere of the paper.

Page 1. Line 32. Please, review this sentence

Page 3. Line 106. Please, specify the acronyms ArmPWV and PWV

Page 3. Line 118. What was the approach that the authors  used in triglycerides analysis?

Page 3. Lines 118 – 124. “clinical laboratory using conventional assays”. Please, specify the analytical techniques

I am not convinced with the following results:

Page 5. Lines 167 – 170. No significant differences was noted in body fat%, muscle mass among the 3 groups. I am not convinced with this suggestion when I see the values for this variables with CON and URT groups (23.1 ± 2.5 and 18.5 ± 2.4; 25.0 ± 1.3 and 30.2 ± 3.4)

Page 5. Lines 171 – 172. “The URT group had a significant improvement in muscle mass and arm LBM (p < 0.01)”. I cannot appreciate these significant differences, seeing the standard deviation and mean values (30.2 ± 3.4 and 31.0 ± 0.4; 2.6 ± 0.3 and 2.8 ± 0.4)

Page 5. Line 173. “However, the LRT group showed no improvement in muscle mass and lega LBM”. However, the differences between variables are similar to those reported in the above consideration (8.0 ± 0.6 and 8.1 ± 0.6).

Did the authors make an unconscious mistake when performing the statistical treatment?

Table 2 is not mentioned in the text

Page 6. Lines 181. What about control group? Was Aix significant reduced?

Lines 187 – 193 are more related with a fail experiment

Lines 207 – 214. This paragraph is more appropriate for Introduction section than in a Discussion section. Please, delete of this part

Lines 215 – 233. This paragraph is not related with the title of the paper. It discusses the improvement of participants’ muscle mass but there is not anything about artery stiffness

Author Response

We would like to thank you for your time and effort to provide the thoughtful comments and suggestions. We believe that the revised manuscript is much improved following the suggested revisions.

I am not convinced with some results show of this study, mainly those reported from line 165 to line 186. Furthermore, the number of participants is very low (only 24). The results are not significant and they could only be showed as a pilot experiment. If the paper were accepted, this aspect should show in somewhere of the paper.

Thank you for your thoughtful comments. To increase the clarity of the content, some information has been corrected and added. Please refer to the highlighted words and statistical numbers from line 165 to line 167 in the result. According to your suggestion, the title of the manuscript has been changed.

Page 1. Line 32. Please, review this sentence

We have reviewed line 32 in the introduction, but we could not find any problems with grammar and content.

Page 3. Line 106. Please, specify the acronyms ArmPWV and PWV

This information has been added to line 106 ~ 107. Please refer to the highlighted sentences in the materials and methods. Thanks for your comments.

Page 3. Line 118. What was the approach that the authors  used in triglycerides analysis?

Page 3. Lines 118 – 124. “clinical laboratory using conventional assays”. Please, specify the analytical techniques

Thank you for your sharp point. We have added specific triglycerides and catecholamine assays into the blood chemistry section. In addition, we have deleted an unnecessary sentence including “conventional assays” in the section. Please refer to the highlighted words and phrases.

Page 5. Lines 167 – 170. No significant differences was noted in body fat%, muscle mass among the 3 groups. I am not convinced with this suggestion when I see the values for this variables with CON and URT groups (23.1 ± 2.5 and 18.5 ± 2.4; 25.0 ± 1.3 and 30.2 ± 3.4)

The data present mean and standard error, not standard deviation. I guess that this may cause you confusion. Regarding fat%, F-value was 1.68 and ANOVA P-value was 0.21 (between df was 2, within df was 21). F-value and P-value of muscle mass were 1.121 and 0.345 respectively (between df was 2, within df was 21).

Page 5. Lines 171 – 172. “The URT group had a significant improvement in muscle mass and arm LBM (p < 0.01)”. I cannot appreciate these significant differences, seeing the standard deviation and mean values (30.2 ± 3.4 and 31.0 ± 0.4; 2.6 ± 0.3 and 2.8 ± 0.4)

The data present mean and standard error, not standard deviation. I guess that this may cause you confusion. We found a typo in the standard error of post-training muscle mass mean. We have corrected it. Please refer to the highlighted number in table 1. For your information, the T-value and P-value of the URT muscle mass paired t-test was -4.220 and 0.004 respectively (df was 7). The T-value and P-value of the URT arm LBM was -3.516 and 0.01 respectively (df was 7). Many thanks for your sharp point.

Page 5. Line 173. “However, the LRT group showed no improvement in muscle mass and lega LBM”. However, the differences between variables are similar to those reported in the above consideration (8.0 ± 0.6 and 8.1 ± 0.6).

The data present mean and standard error, not standard deviation. I guess that this may cause you confusion. For your information, the T-value and P-value of the LRT muscle mass paired t-test was -1.770 and 0.12 respectively (df was 7). The T-value and P-value of the LRT leg LBM was -0.613 and 0.559 respectively (df was 7).

Did the authors make an unconscious mistake when performing the statistical treatment?

We double-checked our statistical process, and corrected some typos in numbers. Thank you for catching some mistakes. 

Table 2 is not mentioned in the text

We have mentioned table 2 between line 175 and 177.

Page 6. Lines 181. What about control group? Was Aix significant reduced?

Thanks for your question. AIx in the CON group was reduced in response to 8-week intervention, but did not reach to the established statistical significance. The T-value and P-value of paired t-test were 2.182 and 0.065 respectively (df was 7). While rechecking the AIx number in CON group, errors in Figure 1 were found and corrected. Many thanks to you again.

Lines 187 – 193 are more related with a fail experiment

With both URT and LRT groups, plasma catecholamine concentration in CON group was also significantly decreased. We tried to find if there is an unintended mistake for handling and analyzing our blood samples, but couldn’t find any problem. Thus, we concluded that the changes in plasma catecholamine level would be associated with seasonal variation of them. We mentioned it in the last paragraph of the discussion section.  

Lines 207 – 214. This paragraph is more appropriate for Introduction section than in a Discussion section. Please, delete of this part

We agree with your suggestions for improvement. This part in the discussion section was deleted.

Lines 215 – 233. This paragraph is not related with the title of the paper. It discusses the improvement of participants’ muscle mass but there is not anything about artery stiffness

We agree with your comment for improvement. The paragraph was deleted. Thanks much for your sharp point.

Reviewer 2 Report

This manuscript is well-written but lacks scientific soundness in the way in which the data was analyzed. The authors report analyzing the data with dependent t-tests for each of the three groups for the pre-post design. I would highly suggest that a repeated measures ANOVA would be more appropriate in the analysis of the data.

Major comments:

  1. The introduction is sound and lays the groundwork for why the study was completed; however, a specific hypothesis statement should be included at the end of the introduction so the reader knows what the authors are hypothesizing with the study design.
  2. The number of participants in the study needs to be clarified. It is currently stated that "Twenty-five young adults (9 men and 15 women...)" were recruited - the 9 men and 15 women only adds up to 24. It is also confusing whether the study ended with 23 participants or 24 participants as it is mentioned that 1 participant dropped out. Please clarify this.
  3. Was any physical activity tracking done in the control group? Please clarify.
  4. Line 86 mentions a 'validated' method used for arterial stiffness - please provide a reference for this validation. Also, please provide a reference for the reliable measure of AoPWV mentioned in line 94-95.
  5. For Table 2 - what are the strength measures units in? Also 'seated row' is spelled incorrectly in the table.
  6. Figure 2 - it is hard to see the post measurements in the figure provided. Please improve this by using hatched boxes for the post measurement in the figure.
  7.  In the beginning of the discussion, please include a statement that refers back to your original hypothesis statement.
  8. Also in the discussion, please include some of the limitations of the study such as small sample size and changes that may have happened due to seasonal effects in the dependent variables. 

Author Response

We would like to thank you for your time and effort to provide the thoughtful comments and suggestions. We believe that the revised manuscript is much improved following the suggested revisions.

This manuscript is well-written but lacks scientific soundness in the way in which the data was analyzed. The authors report analyzing the data with dependent t-tests for each of the three groups for the pre-post design. I would highly suggest that a repeated measures ANOVA would be more appropriate in the analysis of the data.

Thanks for your suggestions. We also speculated which statistical approach would be better when presenting readers our study results based on research hypotheses. This study was not to compare the differences between upper-limb and lower-limb resistance training effects, but to comprehensively present the effect of resistance training of each limb on central and peripheral arterial stiffness. Therefore, we conducted a paired t-test for each group instead of ANOVA with repeated measures for the major dependent variables.  

The introduction is sound and lays the groundwork for why the study was completed; however, a specific hypothesis statement should be included at the end of the introduction so the reader knows what the authors are hypothesizing with the study design.

Thanks for your opinion. Frankly, I am afraid that the contents would look redundant if I write down both the research purpose and the research hypothesis. In general, thesis and dissertation include both research purposes and hypothesis. However, research papers are often written only for research purposes to pursue brevity. Since the research purpose is derived based on the research hypothesis, we would like to write the research purpose only. We seek your generous understanding.

The number of participants in the study needs to be clarified. It is currently stated that "Twenty-five young adults (9 men and 15 women...)" were recruited - the 9 men and 15 women only adds up to 24. It is also confusing whether the study ended with 23 participants or 24 participants as it is mentioned that 1 participant dropped out. Please clarify this.

Thanks much for your sharp point. We have corrected the number issue. Please refer to the highlighted sentences in the materials and methods.

Was any physical activity tracking done in the control group? Please clarify.

Unfortunately, we did not measure physical activity count by tri-axial accelerometers. We asked the study participants in CON group to maintain their usual lifestyle. When the participants visited our laboratory for post-intervention measures, we confirmed if there were any changes in their dietary patterns or physical activity habits.

Line 86 mentions a 'validated' method used for arterial stiffness - please provide a reference for this validation. Also, please provide a reference for the reliable measure of AoPWV mentioned in line 94-95.

Thanks for your keen feedback. We have added the associated references into the sentences you mentioned. Please refer to the highlighted sentences.

For Table 2 - what are the strength measures units in? Also 'seated row' is spelled incorrectly in the table.

Thanks much for your sharp point. We have added the strength unit and corrected the misspelled word in Table 2. Please refer to the highlighted words in Table 2. 

Figure 2 - it is hard to see the post measurements in the figure provided. Please improve this by using hatched boxes for the post measurement in the figure.

Thanks much for your suggestions. We have modified the figure boxes according to your suggestion.

In the beginning of the discussion, please include a statement that refers back to your original hypothesis statement.

Thanks for your opinion. For the consistency of the content, a sentence that evokes the purpose of the study was written at the beginning of the discussion section. Please refer to the highlighted sentence in the first paragraph of the discussion section.

Also in the discussion, please include some of the limitations of the study such as small sample size and changes that may have happened due to seasonal effects in the dependent variables. 

Thanks much for your thoughtful comments. We have added the study limitations into the last paragraph of the discussion section. Please refer to the highlighted paragraph.

Reviewer 3 Report

This is a small randomized controlled trial, investigating effect of upper and lower  limb resistance training on arterial stiffness in young adults. Authors conclude that upper limb training improves arterial stiffness.

Although generalizability of results is weakened by the small sample size, effects of intervention are supported by diligent methodology.

The study look suitable for publication without further revisions.

Author Response

This is a small randomized controlled trial, investigating effect of upper and lower  limb resistance training on arterial stiffness in young adults. Authors conclude that upper limb training improves arterial stiffness.

Although generalizability of results is weakened by the small sample size, effects of intervention are supported by diligent methodology.

The study look suitable for publication without further revisions.

We deeply appreciate your positive comments.

Round 2

Reviewer 1 Report

I still think that this paper should not be accepted for publication. The authors made basic mistakes. The paper has a poor scientific quality in somer relevant parts

Line 126. Please, provide the brand or manufacturer of enzymatic colorimetric assay

Line 127 - 128. Please, provide the bran or manufacturer of HPLC equipment

Lines 167 - 170; 171 - 172 and 173 of my first revision. Please, review my initial considerations.

According with the authors, they showed the standard error instead standard deviation. However, "the standard error (SE) of a statistic (usually an estimate of a parameter) is the standard deviation of its sampling distribution"  

Author Response

I still think that this paper should not be accepted for publication. The authors made basic mistakes. The paper has a poor scientific quality in somer relevant parts.

I feel very sorry that you are not satisfied with my first responses and the revised manuscript. I think there is a slight difference between you and me in the concept of statistical terms and the way you view statistical results. To try to clear up unnecessary misunderstanding between you and me, I have included all raw data and relevant statistics on the statistic issues you have raised in this response. Hope you are satisfied with my answers and responses to your precious comments and suggestions this time.

Line 126. Please, provide the brand or manufacturer of enzymatic colorimetric assay

Thanks for your precious comments. We have added the detailed information of the indicated assay method to the revised manuscript. Please refer to the highlighted sentences in the materials and methods.

Line 127 - 128. Please, provide the bran or manufacturer of HPLC equipment

Thanks for your precious comments. We have added the detailed information of the indicated equipment to the revised manuscript. Please refer to the highlighted sentences in the materials and methods.

Lines 167 - 170; 171 - 172 and 173 of my first revision. Please, review my initial considerations.

[Initial consideration of the reviewer] Page 5. Lines 167 – 170. No significant differences was noted in body fat%, muscle mass among the 3 groups. I am not convinced with this suggestion when I see the values for this variables with CON and URT groups (23.1 ± 2.5 and 18.5 ± 2.4; 25.0 ± 1.3 and 30.2 ± 3.4)

[1st Response to the reviewer’s comment] The data present mean and standard error, not standard deviation. I guess that this may cause you confusion. Regarding fat%, F-value was 1.68 and ANOVA P-value was 0.21 (between df was 2, within df was 21). F-value and P-value of muscle mass were 1.121 and 0.345 respectively (between df was 2, within df was 21).

[2nd Response to the reviewer’s comment] For your better understanding, I have added my raw data and associated statistic values here. Please review my raw data and related statistic values.

Descriptives

Group

N

Mean

Standard

Deviation

Standard

Error

95% CI for Mean

Min

Max

Lower

Upper

%fat

_pre

CON

8

23.1

7.1188

2.5169

17.15

29.05

10.6

33.5

URT

8

18.45

6.7638

2.3914

12.8

24.11

10.2

25.6

LRT

8

24.287

6.2891

2.2235

19.03

29.55

13.3

30.3

SUM

24

21.946

6.9286

1.4143

19.02

24.87

10.2

33.5

MuscleMass

_Pre

CON

8

25

3.5456

1.2536

1.2536

22.04

19.7

29.5

URT

8

30.163

9.4989

3.3584

3.3584

22.22

19.6

42.7

LRT

8

27.638

6.316

2.233

2.233

22.36

21.1

36.6

SUM

24

27.6

6.9328

1.4151

1.4151

24.67

19.6

42.7

One-way ANOVA

SS

df

MS

F

P-value

%fat

_pre

Between

152.291

2

76.145

1.68

0.21

Within

951.849

21

45.326

Total

1104.14

23

MuscleMass

_Pre

Between

106.623

2

53.311

1.121

0.345

Within

998.838

21

47.564

Total

1105.46

23

ID

Group

Group coding

%BF_pre

MuscleMass_pre

Participant 1

LRT

2

13.3

33.9

Participant 2

LRT

2

23.6

23.6

Participant 3

URT

1

23.6

25.1

Participant 4

LRT

2

30.3

24.2

Participant 5

URT

1

18.1

19.6

Participant 6

URT

1

25.6

23.9

Participant 7

CON

0

33.5

26.8

Participant 8

LRT

2

28.3

21.1

Participant 9

LRT

2

16

36.6

Participant 10

URT

1

10.2

42.1

Participant 11

URT

1

23.5

23.9

Participant 12

URT

1

24.6

24.6

Participant 13

URT

1

11.2

39.4

Participant 14

LRT

2

28.6

23.1

Participant 15

CON

0

18.7

28

Participant 16

CON

0

19.1

25.6

Participant 17

LRT

2

26.7

34.9

Participant 18

CON

0

21.7

20.9

Participant 19

LRT

2

27.5

23.7

Participant 20

CON

0

10.6

27

Participant 21

CON

0

28

22.5

Participant 22

CON

0

25.1

19.7

Participant 23

URT

1

10.8

42.7

Participant 24

CON

0

28.1

29.5

[Initial consideration of the reviewer] Page 5. Lines 171 – 172. “The URT group had a significant improvement in muscle mass and arm LBM (p < 0.01)”. I cannot appreciate these significant differences, seeing the standard deviation and mean values (30.2 ± 3.4 and 31.0 ± 0.4; 2.6 ± 0.3 and 2.8 ± 0.4)

[1st Response to the reviewer’s comment] The data present mean and standard error, not standard deviation. I guess that this may cause you confusion. We found a typo in the standard error of post-training muscle mass mean. We have corrected it. Please refer to the highlighted number in table 1. For your information, the T-value and P-value of the URT muscle mass paired t-test was -4.220 and 0.004 respectively (df was 7). The T-value and P-value of the URT arm LBM was -3.516 and 0.01 respectively (df was 7). Many thanks for your sharp point.

[2nd Response to the reviewer’s comment] For your better understanding, I have added my raw data and associated statistic values here. Please review my raw data and related statistic values.

Descriptives

Mean

N

Standard Deviation

Standard Error

Pair1

Musclemass_pre

30.163

8

9.4989

3.3584

Musclemass_post

31.025

8

9.7388

3.4432

Pair2

ArmLBM_pre

2.64

8

0.9321

0.3295

ArmLBM_post

2.793

8

1.0045

0.3552

Paired Samples Test

Paired Differences

t

df

Sig. (2-tained)

Mean

Standard Deviation

Standard Error Mean

95% CI for the Difference

Lower

Upper

Pair1

Musclemass_pre-Musclemass_post

-0.8625

0.5780

0.2044

-1.3457

-0.3793

-4.220

7

0.004

Pair2

ArmLBM_pre-ArmLBM_post

-0.1531

0.1231

0.0435

-0.2561

-0.0502

-3.516

7

0.010

ID

Group

MuscleMass
_pre

MuscleMass
_post

ArmLBM
_pre

ArmLBM
_post

Participant 3

URT

25.1

24.9

2.2

2.1

Participant 5

URT

19.6

20.1

1.6

1.7

Participant 6

URT

23.9

24.8

2.1

2.2

Participant 10

URT

42.1

43.4

3.9

4.2

Participant 11

URT

23.9

25.6

2.1

2.4

Participant 12

URT

24.6

25.2

2

2.1

Participant 13

URT

39.4

40.3

3.4

3.6

Participant 23

URT

42.7

43.9

3.9

4.1

[Initial consideration of the reviewer] Page 5. Line 173. “However, the LRT group showed no improvement in muscle mass and lega LBM”. However, the differences between variables are similar to those reported in the above consideration (8.0 ± 0.6 and 8.1 ± 0.6).

[1st Response to the reviewer’s comment] The data present mean and standard error, not standard deviation. I guess that this may cause you confusion. For your information, the T-value and P-value of the LRT muscle mass paired t-test was -1.770 and 0.12 respectively (df was 7). The T-value and P-value of the LRT leg LBM was -0.613 and 0.559 respectively (df was 7).

[2nd Response to the reviewer’s comment] For your better understanding, I have added my raw data and associated statistic values here. Please review my raw data and related statistic values.

Descriptives

Mean

N

Standard Deviation

Standard Error

Pair1

Musclemass_pre

27.638

8

6.316

2.233

Musclemass_post

28.088

8

5.9097

2.0894

Pair2

LegLBM_pre

8.044

8

1.6191

0.5724

LegLBM_post

8.087

8

1.6704

0.5906

Paired Samples Test

Paired Differences

t

df

Sig. (2-tained)

Mean

Standard Deviation

Standard Error Mean

95% CI for the Difference

Lower

Upper

Pair1

Musclemass_pre-Musclemass_post

-0.45

0.7191

0.2542

-1.0512

0.1512

-1.77

7

0.12

Pair2

LegLBM_pre-LegLBM_post

-0.0425

0.1961

0.0693

-0.2065

0.1215

-0.613

7

0.559

ID

Group

MuscleMass
_pre

MuscleMass
_post

LegLBM
_pre

LegLBM
_post

Participant 1

LRT

33.9

33.8

9.9

10.3

Participant 2

LRT

23.6

25

7.2

7.3

Participant 4

LRT

24.2

25.2

7

7.1

Participant 8

LRT

21.1

21.2

6.5

6.5

Participant 9

LRT

36.6

37

10.2

10

Participant 14

LRT

23.1

23.6

6.8

6.6

Participant 17

LRT

34.9

34.1

9.9

9.9

Participant 19

LRT

23.7

24.8

6.9

7

According with the authors, they showed the standard error instead standard deviation. However, "the standard error (SE) of a statistic (usually an estimate of a parameter) is the standard deviation of its sampling distribution"

Thanks for your explanation. In this study, the standard error (SE) means the standard deviation divided by the square root n. Unfortunately, I am not a statistician. Thus, please let me know if I misunderstand or misuse the concepts of SE and SD in this study when I present my study results based on SPSS statistical analyses.

Reviewer 2 Report

In the updated manuscript none of the issues identified in the previous review have been addressed. This is disappointing. Please refer back to the original review and update the manuscript accordingly.

Author Response

I did my best to sincerely write and upload my responses to your review comments and suggestions. However, I am sorry that you did not receive my responses. If I do not respond to even one reviewer, the journal submission system will not allow my revised manuscript to be uploaded. Thank you for your patience and generous understanding, and once again send and upload my responses to your comments and suggestions. Hope you confirm it this time. Below is my sincere responses to your review comments and suggestions I uploaded yesterday.

=================================================================================

We would like to thank you for your time and effort to provide the thoughtful comments and suggestions. We believe that the revised manuscript is much improved following the suggested revisions.

This manuscript is well-written but lacks scientific soundness in the way in which the data was analyzed. The authors report analyzing the data with dependent t-tests for each of the three groups for the pre-post design. I would highly suggest that a repeated measures ANOVA would be more appropriate in the analysis of the data.

Thanks for your suggestions. We also speculated which statistical approach would be better when presenting readers our study results based on research hypotheses. This study was not to compare the differences between upper-limb and lower-limb resistance training effects, but to comprehensively present the effect of resistance training of each limb on central and peripheral arterial stiffness. Therefore, we conducted a paired t-test for each group instead of ANOVA with repeated measures for the major dependent variables.  

The introduction is sound and lays the groundwork for why the study was completed; however, a specific hypothesis statement should be included at the end of the introduction so the reader knows what the authors are hypothesizing with the study design.

Thanks for your opinion. Frankly, I am afraid that the contents would look redundant if I write down both the research purpose and the research hypothesis. In general, thesis and dissertation include both research purposes and hypothesis. However, research papers are often written only for research purposes to pursue brevity. Since the research purpose is derived based on the research hypothesis, we would like to write the research purpose only. We seek your generous understanding.

The number of participants in the study needs to be clarified. It is currently stated that "Twenty-five young adults (9 men and 15 women...)" were recruited - the 9 men and 15 women only adds up to 24. It is also confusing whether the study ended with 23 participants or 24 participants as it is mentioned that 1 participant dropped out. Please clarify this.

Thanks much for your sharp point. We have corrected the number issue. Please refer to the highlighted sentences in the materials and methods.

Was any physical activity tracking done in the control group? Please clarify.

Unfortunately, we did not measure physical activity count by tri-axial accelerometers. We asked the study participants in CON group to maintain their usual lifestyle. When the participants visited our laboratory for post-intervention measures, we confirmed if there were any changes in their dietary patterns or physical activity habits.

Line 86 mentions a 'validated' method used for arterial stiffness - please provide a reference for this validation. Also, please provide a reference for the reliable measure of AoPWV mentioned in line 94-95.

Thanks for your keen feedback. We have added the associated references into the sentences you mentioned. Please refer to the highlighted sentences.

For Table 2 - what are the strength measures units in? Also 'seated row' is spelled incorrectly in the table.

Thanks much for your sharp point. We have added the strength unit and corrected the misspelled word in Table 2. Please refer to the highlighted words in Table 2. 

Figure 2 - it is hard to see the post measurements in the figure provided. Please improve this by using hatched boxes for the post measurement in the figure.

Thanks much for your suggestions. We have modified the figure boxes according to your suggestion.

In the beginning of the discussion, please include a statement that refers back to your original hypothesis statement.

Thanks for your opinion. For the consistency of the content, a sentence that evokes the purpose of the study was written at the beginning of the discussion section. Please refer to the highlighted sentence in the first paragraph of the discussion section.

Also in the discussion, please include some of the limitations of the study such as small sample size and changes that may have happened due to seasonal effects in the dependent variables. 

Thanks much for your thoughtful comments. We have added the study limitations into the last paragraph of the discussion section. Please refer to the highlighted paragraph.

Round 3

Reviewer 1 Report

I have already revised this manuscript twice and my opinion remains the same.

Best regards.

Reviewer 2 Report

Thank you for addressing my comments and queries. I have no further feedback for the authors.